# Acoustic modes in M67 cluster stars trace deepening convective envelopes

Claudia Reyes[1,2 ✉], Dennis Stello[1,3], Joel Ong[4], Christopher Lindsay[5], Marc Hon[4,6] & Timothy R. Bedding[3]

Acoustic oscillations in stars are sensitive to stellar interiors[1]. Frequency differences between overtone modes—large separations—probe stellar density[2], whereas differences between low-degree modes—small separations—probe the sound-speed gradient in the energy-generating core of main-sequence Sun-like stars[3], and hence their ages. At later phases of stellar evolution, characterized by inert cores, small separations are believed to lose much of their power to probe deep interiors and become proportional to large separations[4,5]. Here we present evidence of a rapidly evolving convective zone as stars evolve from the subgiant phase into red giants. By measuring acoustic oscillations in 27 stars from the open cluster M67, we observe deviations of proportionality between small and large separations, which are caused by the influence of the bottom of the convective envelope. These deviations become apparent as the convective envelope penetrates deep into the star during subgiant and red giant evolutions, eventually entering an ultradeep regime that leads to the red-giant-branch luminosity bump. The tight sequence of cluster stars, free of large spreads in ages and fundamental properties, is essential for revealing the connection between the observed small separations and the chemical discontinuities occurring at the bottom of the convective envelope. We use this sequence to show that combining large and small separations can improve estimations of the masses and ages of field stars well after the main sequence.

The oscillation spectra of Sun-like stars and their evolved counterparts, subgiants and red giants originate from resonating acoustic pressure (*p*) waves excited by surface convection (Fig. 1a). The *p*-modes of spherical degree $\ell = 0$ travel radially through the star and are reflected towards the core at the stellar surface. The non-radial *p*-modes (degree $\ell \geq 1$) are refracted back to the surface and, therefore, are confined between an inner turning point and the surface. The radial coordinate of the inner turning point is a function of the spherical degree of the mode and the temperature gradient in the core.

Small separations refer to the frequency differences between the modes of degrees $\ell$ and $\ell + 2$, of consecutive order *n*. Owing to the low visibility of the modes of degree $\ell \geq 3$ (ref. 6), we focus on $\delta\nu_{0,2}$, the small separation between modes of degrees $\ell = 0$ and $\ell = 2$. This separation is typically determined from models as a weighted average[4] of individual $\ell = 0, 2$ pairs, with weights determined by the frequency distance between the mode of degree $\ell = 0$ and order *n*, and the frequency of maximum oscillation power, $\nu_{max}$. Asymptotic analysis[7–9] yields the approximate expression $\delta\nu_{0,2} \simeq -\frac{3}{\nu T} \int_0^R \frac{dc_s}{dr} \frac{dr}{r}$ for a given frequency $\nu$, where *T* is the acoustic radius, *R* is the radius, $c_s$ is the speed of sound and *r* is the radial coordinate. In main-sequence stars, this reduces to $\delta\nu_{0,2} \propto \sqrt{1/\mu}$, where $\mu$ is the mean molecular weight[10]. Therefore, $\delta\nu_{0,2}$ rapidly changes as the star burns hydrogen into helium. In main-sequence stars, small separations are a good indicator of evolutionary state[11], and hence age. Notably, the accuracy of this asymptotic approximation rapidly deteriorates as $\nu_{max}$ decreases and the star becomes more centrally condensed during its evolution[8]. Because of the lack of a suitable analytical expression for $\delta\nu_{0,2}$ in subgiants and red giants, the relationship between small separations and the interior structure of stars is not fully understood after the main sequence. A model-based interpretation has also remained out of reach because of the mixed nature of non-radial modes in subgiants and red giants, which result from the coupling between *p*-waves and *g*-waves (gravity waves trapped in the core)[12,13]. This coupling produces irregular mode patterns and scatter in small separations[4], making it difficult to obtain useful model predictions of small separations. Although the coupling weakens for late red-giant-branch stars[14], $\delta\nu_{0,2}$ becomes nearly proportional to the large frequency separations, $\Delta\nu$, limiting the information it provides about the star[15–17].

## Small frequency separations follow hydrogen fusion zones

Figure 2a shows the known critical stellar evolution points in the Hertzsprung–Russell diagram (A, B, C, D and G), which are associated with structural changes in the hydrogen-burning regions seen in the Kippenhahn diagram (Fig. 2b). By using modelled pure *p*-modes[18] isolated

[1]School of Physics, University of New South Wales, Sydney, New South Wales, Australia. [2]Research School of Astronomy and Astrophysics, Australian National University, Canberra, Australian Capital Territory, Australia. [3]Sydney Institute for Astronomy (SIfA), School of Physics, University of Sydney, Sydney, New South Wales, Australia. [4]Institute for Astronomy, University of Hawaii, Honolulu, HI, USA. [5]Department of Astronomy, Yale University, New Haven, CT, USA. [6]Department of Physics and Kavli Institute for Astrophysics and Space Research, Massachusetts Institute of Technology, Cambridge, MA, USA. ✉e-mail: c.reyes_saez@unsw.edu.au

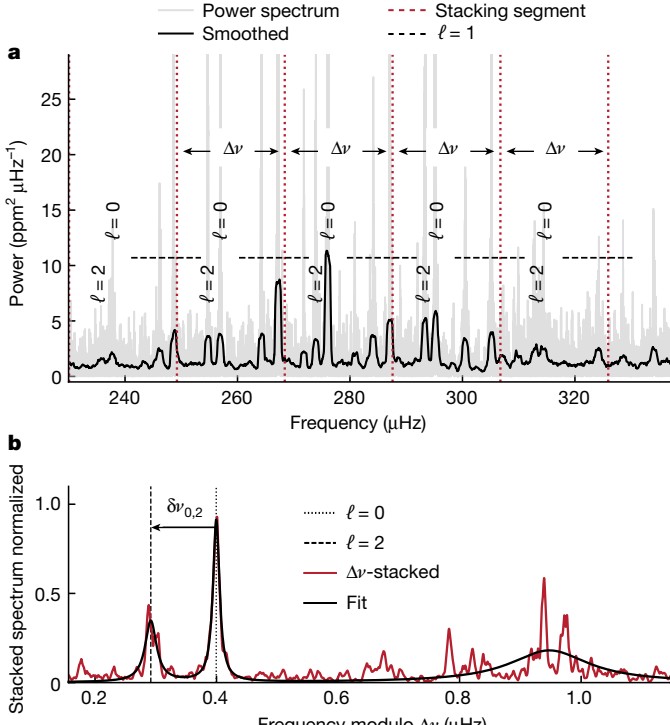

**Fig. 1 | Oscillations in the red giant EPIC 211409560 in the open cluster M67. a**, Region of the oscillation power spectrum (grey) centred around the frequency of maximum oscillation power. A slightly smoothed version of the spectrum is presented in black for clarity. The $\ell = 0$ and $\ell = 2$ modes are annotated, and the approximate ranges occupied by the $\ell = 1$ modes are indicated by horizontal dashed lines. The vertical dotted red lines indicate the segments used to stack the spectrum (frequency modulo $\Delta\nu$ ($\mu$Hz) in **b**). **b**, Stacked spectrum (red) and the sum of three Lorentzian functions (black) fitted to the stacked spectrum. The dotted and dashed black vertical lines mark the centres of the fitted Lorentzian profiles to the $\ell = 0$ and $\ell = 2$ modes, respectively. We measure the small frequency separation $\delta\nu_{0,2}$ as the distance between these centres, as indicated by the arrow.

from the inner $g$-mode cavity, we produce evolution sequences of small versus large frequency separations, in so-called C–D diagrams[11], showing no mixed-mode-induced scatter that would otherwise strongly distort the sequences[4]. The smooth C–D diagrams (Fig. 2c) now also show the imprint of all these critical points. Importantly, we see a new morphological feature in the C–D diagram, which we call the plateau, bracketed by E and F. This feature, which has no counterpart in the other diagrams, appears during thin-shell burning as a temporary stalling in small frequency separations, while $\Delta\nu$ continues to decrease.

## The M67 plateau feature

The near-solar-metallicity open cluster M67 (NGC 2682) presents a unique opportunity to investigate the nature of the plateau feature. This cluster has a rich subgiant and red giant population, which has been the target of attempted seismic studies for decades[19]. Recent work includes a study of its giants[20] using Kepler/K2 data[21], which we also use in this study.

We analysed spectra from 27 shell-hydrogen-burning stars (Extended Data Table 1) and determined their $\delta\nu_{0,2}$ using a method that mitigates the influence of mixed modes (Methods), shown in Fig. 1b. Figure 3a shows the M67 C–D diagram, in which the evolutionary state goes from subgiants (right) to red giants (left) as indicated by the black arrow. The models used to produce this pure $p$-mode C–D diagram correspond to a 3.95-Gyr theoretical isochrone specifically

designed to provide the closest fit to M67 photometry to date[22]. The post-main-sequence section shown in Fig. 3a represents the evolved segment of the complete isochrone shown in Fig. 3b, and corresponds to stellar models in the mass range 1.30–1.37$M_\odot$. In the observations, shown in black circles, we detect the plateau where $\delta\nu_{0,2}$ remains almost constant in the well-populated evolutionary locus of stars at $\Delta\nu$ between about 17–22 $\mu$Hz, indicated with a grey box in Fig. 3a. This feature, which is closely reproduced by the models, is evident in the data, and it probably remained undiscovered until now only because of the lack of a uniform sample of stars with similar fundamental properties (and hence, no intrinsic star-to-star scatter) needed to reveal it.

As indicated in the Gaia colour–magnitude diagram in Fig. 3c with a grey box, the plateau occurs in stars as they ascend the red giant branch. In these stars, the electron-degenerate core contracts as it grows in mass, fed by the ashes of the hydrogen-burning shell. As per the mirror principle, the envelope expands and cools down, with the convective region deepening because of the increasing photospheric opacity of the cooling envelope[23,24].

We investigated the relative contributions of the $\ell = 0$ and $\ell = 2$ modes to the plateau in $\delta\nu_{0,2}$ using the concept of internal phase shifts[3] $\phi_\ell$, given that $\delta\nu_{0,2} \sim \frac{\Delta\nu}{\pi}(\phi_2(\nu) - \phi_0(\nu))$ (refs. 25,26). This is shown in Extended Data Fig. 1, in which the frequency range of the plateau in our observational C–D diagram is highlighted by the red sections. Within this range, the evolution of the quadrupole-mode inner phase $\phi_2$ can be seen to progress smoothly, whereas it is the evolution of the radial-mode inner phase $\phi_0$ that exhibits a local minimum, thereby producing the observed plateau. Thus, we conclude that the plateau in the observed C–D diagram probes stellar structural features lying near the centre of the star, at which only radial ($\ell = 0$) $p$-modes reach, beyond the inner turning point of $\ell = 2$ modes.

## The lower boundary of the convective envelope

We find that the observed plateau can be traced to the evolution of the lower boundary of the convective envelope. As the envelope expands and cools down, this lower boundary extends ever deeper into the stellar interior (Figs. 2b and 3d) as more efficient energy transport mechanisms are required deeper in the star. Large density and sound-speed gradients are known to exist at these boundaries because of differing chemical compositions on either side, as shown in Extended Data Fig. 2. These gradients produce 'acoustic glitches'[1,27–29], imparting frequency differences $\delta\nu_{glitch}$ compared with the mode frequencies of a smooth stellar structure with weaker gradients. By writing the difference in density between the actual structure and such a smooth model as $\delta\rho$, the acoustic glitch signal may be described through expressions of the form $\delta\nu_{\text{glitch},i} \sim \sum_q \int K_{q,i}\delta q(m)\mathrm{d}m$, where $m(r) = \int_0^r 4\pi r'^2\rho(r')\mathrm{d}r'$ is a mass coordinate and $K_{q,i}$ is a sensitivity kernel associated with the quantity $q$ for the $i$th mode. The average effect of $q$ on the radial-mode frequencies may then be examined by inspecting the averaged kernel $\langle K_q \rangle$ over radial modes near $\nu_{\max}$ (Methods). For illustration, we take the amplitude of the density kernel $\langle K_{\rho,c_s^2} \rangle$ (shown by the background colouring of Fig. 3d) along the position of the mixing boundary (solid black line) in mass coordinates and we colour-code the seismic isochrone accordingly. In Fig. 3a, the C–D diagram seems to be modulated by the amplitude of the kernel at the mixing boundary. Furthermore, the plateau in the M67 C–D diagram occurs when the mixing boundary sweeps over one of the extreme points of this sensitivity kernel denoted by the darkest blue in Fig. 3.

To verify the connection between $\delta\nu_{0,2}$ and the bottom of the convection zone, we examine how the former changes when we vary the latter in our stellar modelling by altering the amount of convective boundary mixing in our computational treatment of stellar structure and evolution. In practice, we perform parameterization of this convective

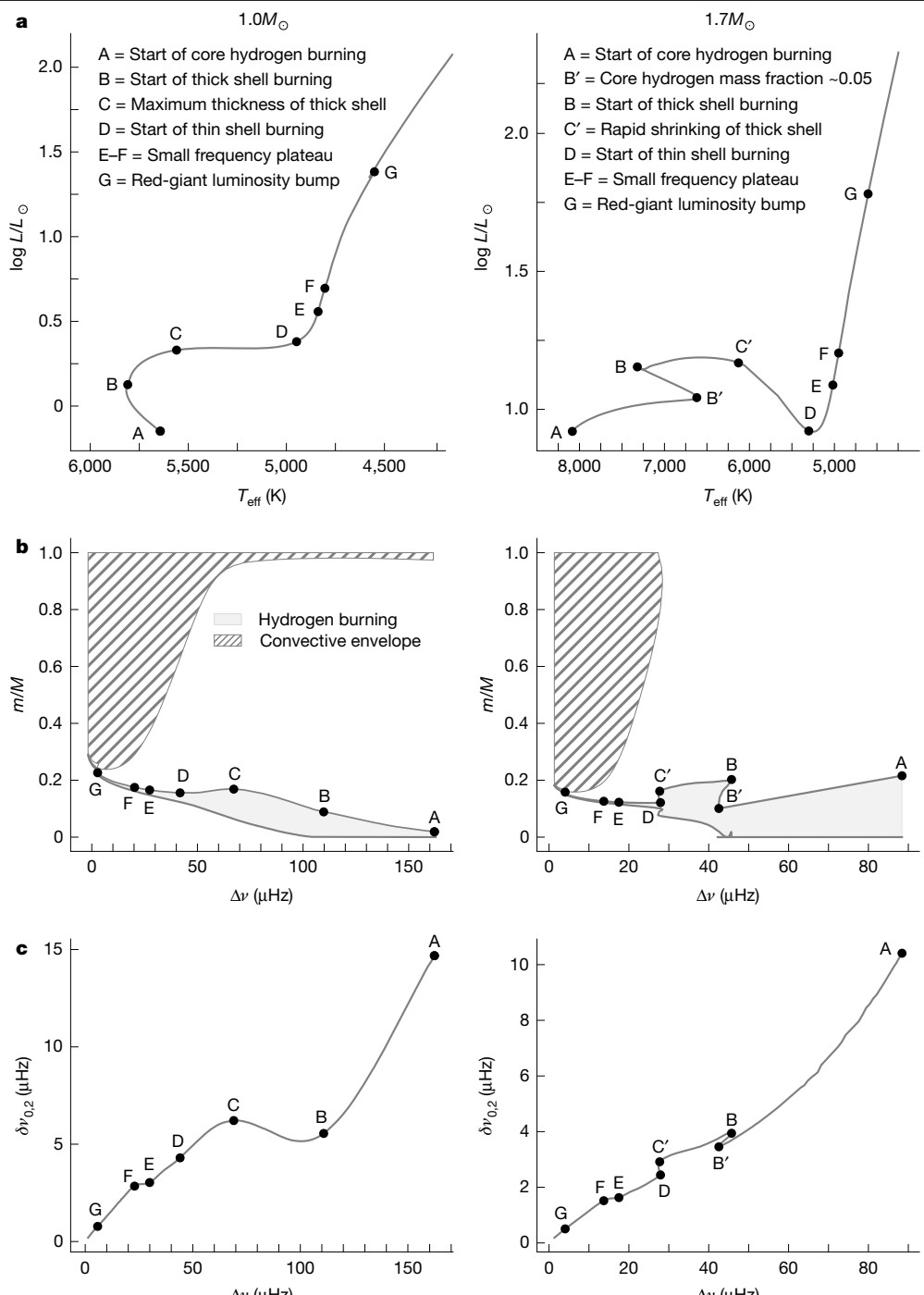

**Fig. 2 | Stellar tracks of solar metallicity. a–c**, Hertzsprung–Russell (**a**), frequency–Kippenhahn (**b**) and C–D (**c**) diagrams of a 1.0$M_\odot$ star (main-sequence radiative core, left) and a 1.7$M_\odot$ star (main sequence convective core, right). The black circle indicating point G, corresponding to the red-giant luminosity bump, is covering the short-lived fluctuation in all diagrams in which the curves temporarily revert directions. In **b**, regions in which nuclear burning produces more than 10 erg g$^{-1}$ s$^{-1}$ are shown in light grey, and envelope convective regions are hatched.

boundary mixing as convective overshooting, in which convective motions extend beyond the nominal convective boundaries because of the inherent momentum of convective plumes[30], extending mixing regions, and thus relocating them relative to the regions of convective stability. In Fig. 3e, we show two variants of the main isochrone: one based on models with no envelope overshoot and the other with twice the extent of overshoot compared with the solar-calibrated overshoot factor[31] of the adopted main isochrone. When we use models with more overshooting, the mixing boundary extends deeper into the radiative region compared with models with less or no overshooting.

This means that the boundary will reach the critical kernel region earlier, and hence, we should see that with more overshooting, the deviation from proportionality occurs earlier in the evolution. Conversely, with no envelope overshoot, the boundary takes longer to reach the same stellar depth, and we should see that the deviation from proportionality occurs later. Figure 3e confirms our predictions and further shows that with more overshooting, the plateau from M67 no longer presents a plateau, but a local maximum in $\delta\nu_{0,2}$ that peaks at about 20.5 μHz. With no overshooting, the feature is less prominent and has an inflection point at $\delta\nu_{0,2} \approx 18$ μHz. A new theoretical expression

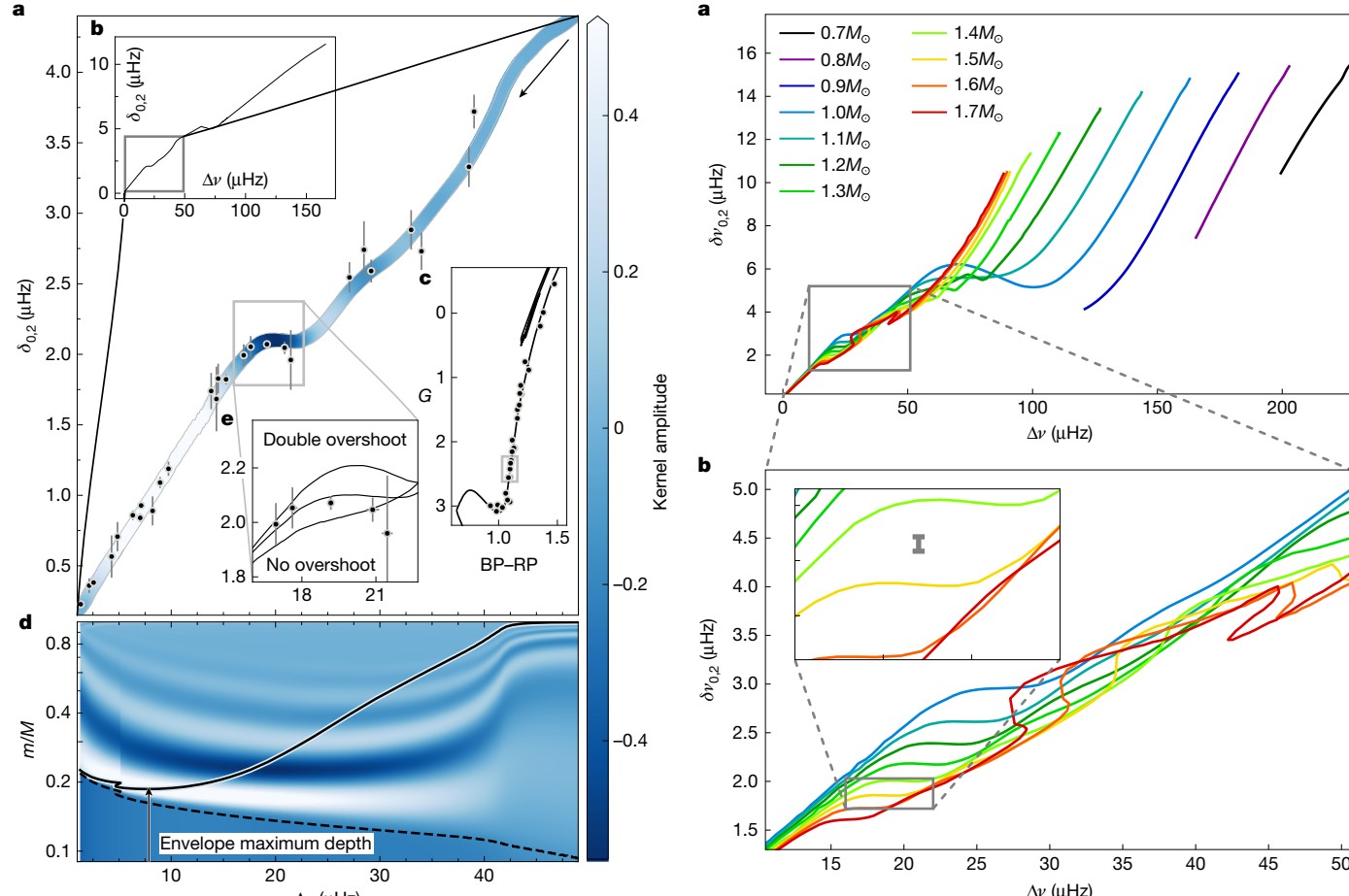

**Fig. 3 | Asteroseismic C–D diagram for subgiants and giants in the open cluster M67. a**, Observed values of the large and small frequency separations for the M67 sample (horizontal axis is shared with **d**). In most cases, the $\Delta\nu$ error bars are smaller than the symbols, indicating that $\Delta\nu$ uncertainties are negligible in this context. The $\delta\nu_{0,2}$ error bars were obtained as detailed in the Methods. The coloured curve represents a theoretical isochrone, colour-coded according to the amplitude of the averaged radial-mode density kernel at the bottom of the convection zone, as shown in **d**. **b**, The theoretical C–D diagram corresponding to our sample of subgiants and red giants (in the grey box) in the context of the full C–D diagram, including the main sequence. **c**, The M67 sample and isochrone plotted in the Gaia colour–magnitude diagram, with a grey box indicating the region corresponding to the grey box in **a**. **d**, $\Delta\nu$ and mass coordinates of the isochrone from **a**. The solid line indicates the bottom of the envelope convection zone, which reaches greater stellar depths as stars evolve from right to left; the dashed line marks the centre of the burning shell and the arrow points to the maximum depth of the envelope. Mass coordinates are colour-coded according to kernel amplitude. **e**, Close-up of the plateau feature of the M67 isochrone (central curve) compared with the same isochrone with double the envelope overshoot (top curve) and with no envelope overshoot (bottom curve). BP, blue photometer; RP, red photometer.

for the small separations in red giants is necessary to fully explain the link between the prominence of the plateau feature and overshooting. However, by comparing with the data, we can say that the adopted solar-calibrated overshoot factor[31] accurately predicts the correct convection zone depth in near-solar metallicity stars such as those in the M67 cluster. The location of the red-giant-branch luminosity bump[32] and the evolutionary behaviour of the $\ell = 1$ mixed modes[33] near the luminosity bump are also dependent on the amount of overshooting at the bottom of the convection zone. Combined with these other indicators of convective envelope depth, the amount of envelope overshooting can now be analysed at several locations along the red giant branch,

**Fig. 4 | The C–D diagram of a sequence of solar metallicity stellar tracks. a**, Stellar tracks between $0.7M_\odot$ and $1.7M_\odot$ starting at the beginning of core hydrogen burning and ending just before the helium flash or at a stellar age of $12 \times 10^9$ years. **b**, The grey box in **a** shown in detail, in which the plateau features are discernible at all masses shown. The inset shows a typical $\delta\nu_{0,2}$ uncertainty for a star between $1.4M_\odot$ and $1.5M_\odot$ observed by Kepler, while typical Kepler $\Delta\nu$ uncertainties are negligible in this context.

because the plateau feature in $\delta\nu_{0,2}$ can sample the amount of envelope overshoot substantially before the luminosity bump.

## Mass dependence of plateau frequencies

Our models also show that small separations behave similarly in other low-mass stars. Therefore, the plateau feature first observed in M67 provides a new diagnostic tool for determining the stellar properties of field stars.

Figure 4 shows a C–D diagram derived from stellar tracks in the range of $0.8$–$1.6M_\odot$, with solar metallicity, starting at the beginning of core hydrogen burning to just before core helium burning, or until a stellar age of $12 \times 10^9$ years, whichever is first. As before, we achieved this level of detail by calculating modelled frequencies using only pure $p$-modes[18]. A plateau feature is well-defined in all tracks shown and appears at values of $\delta\nu_{0,2}$ that are specific for each track. Therefore, by placing observations of $\delta\nu_{0,2}$ and $\Delta\nu$ on grids built from these models, this new diagnostic tool could be used to accurately estimate the masses of field red giants. This is shown in Fig. 4b (inset), which shows typical Kepler $\delta\nu_{0,2}$ uncertainties—and negligible $\Delta\nu$ uncertainties—(Methods), between the plateaus of the $1.4M_\odot$ and $1.5M_\odot$ tracks. Further details on the metallicity dependence of the plateau frequencies are provided in Extended Data Figs. 3 and 4. Outside the $\Delta\nu$ range of the plateau, towards

more evolved giants, the diagnostic power of the post-main-sequence C–D diagram is diminished because the tracks converge.

## M67 unlocks plateau frequencies as stellar probes

As the first single stellar population with clear measurements of small frequency separations across the subgiant and red giant branches, M67 reveals that the depths reached by convective envelopes lead to measurable effects in the small-separation frequencies, which reveals itself as a plateau in the C–D diagram. At the end of the plateau, the convective envelope enters its ultradeep regime, beginning when roughly 80% of the mass of the star is undergoing convection. This fraction continues to increase until the convective envelope reaches its maximum depth. The envelope then retreats, leaving behind a chemical discontinuity imprint. Eventually, this discontinuity is erased as the shell burns through it in what is known as the red giant luminosity bump—until now thought to be the only observational evidence of the depth reached by the convective envelope. The well-defined distribution of plateau frequencies according to mass, combined with the strong mass–age relation for giants[34] and the relative ease of measuring small separations, also make this feature interesting for age-dating red giants in the field and, hence, for mapping the chronology of the Milky Way merger events[35].

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

# Methods

We describe here the seismic analysis of 27 M67 stars observed by the K2 mission, the stellar models we developed for the cluster and the theory behind phase shifts and kernels.

## Oscillation spectra and seismic characterization

**M67 K2 data.** We downloaded all the available K2 light curves of M67 cluster members[22] from the Mikulski Archive for Space Telescopes (https://archive.stsci.edu), corrected them[20] and calculated their power density spectrum[36]. We obtained the initial values of $v_{max}$ and $\Delta v$ and background properties using pysyd (ref. 37), which is a Python implementation of the SYD[38] pipeline. First, we calculated the $\Delta v$-stacked spectrum by averaging the four $\Delta v$-wide segments closest to $v_{max}$ of the background-subtracted power spectrum. We then fitted the sum of three Lorentzian functions to this stacked spectrum as in Fig. 1b. This spectrum stacking method boosts the signal-to-noise ratio of $\ell = 0$ and $\ell = 2$ modes while also reducing the impact of mixed modes in the final $\delta v_{0,2}$ measurements. To quantify the uncertainty in $\delta v_{0,2}$, we calculated the fractional differences between the widths and signal-to-noise ratios of the Lorentzians fitted for $\ell = 0$ and $\ell = 2$ modes. After summing the squares of these differences and taking the square root of the sum, we incorporated the resulting value with the propagated uncertainties from the individual fits. We found that this combined uncertainty is sensitive to missing modes and unusually high signal modes, which are the primary contributors to inaccuracies in $\delta v_{0,2}$ in relatively short lightcurve data, such as K2 data. To refine pipeline $\Delta v$ determinations, we sought the $\Delta v$ value that maximized the height and minimized the width of the Lorentzian fit to the $\ell = 0$ peak. Finally, we rejected stars whose $\Delta v$-stacked $\ell = 0$ or $\ell = 2$ peaks had a signal-to-noise ratio lower than 3, resulting in a sample of 27 stars. The values are presented in Extended Data Table 1.

**Kepler large and small-separation uncertainties.** We calculated an average fractional uncertainty of 0.05% in $\Delta v$ and 0.7% in $\delta v_{0,2}$ based on measurements of 188 Kepler red giants within the range $15 < \Delta v < 20\ \mu Hz$ (ref. 39).

## Stellar models

**Tracks and profiles.** To generate the isochrone models with no (or double) envelope overshoot from Fig. 3e, we adapted models from the M67 isochrone[22]. We note that this isochrone used a mass-dependent core overshoot and a fixed solar-calibrated envelope overshoot, both using the exponential overshoot scheme, and H and He content as documented[22]. We adapted the same models to generate the 0.8–1.6$M_\odot$ tracks from Fig. 4 and Extended Data Figs. 3 and 4, except that the tracks shown in these figures use the solar H and He fractions from Asplund 2009 as reference.

**Radial $\ell = 0$ and non-radial $\ell = 2$ $p$-mode frequencies.** We calculated adiabatic frequencies from structure profiles using the oscillations code GYRE v.6.0.1 (ref. 40). To obtain the smooth sequence of the C−D diagram, we required mode frequencies free of any $g$-mode quality. Because only $\ell = 0$ modes are intrinsically independent of any influence from $g$-modes, we applied a formalism based on semi-analytic expressions for the isolation of modes[18] to compute $\ell = 2$ pure $p$-modes. The pulsation equations are decomposed into a pure $p$-mode wave operator and a remainder term from the radiative interior. The eigenvalues of the former are solved, and pure $p$-mode frequencies are recovered by applying perturbation theory to the latter.

**Modelled seismic data.** Surface corrections are required[41,42] to help minimize the impact of poor modelling of the outer layers in one-dimensional stellar evolution codes before we can compare the models to observed frequencies. For all models, we used a smooth form of surface correction that follows the corrections to radial modes of individual stars in our sample[41]. It is a reasonable approach to apply the same surface offset for both $\ell = 0$ and $\ell = 2$ modes, given that we exclusively work with pure $p$-modes. To obtain $\Delta v$ for all the selected models, we weighted the $\ell = 0$ frequencies by a Gaussian window of width $0.25v_{max}$, centred on $v_{max}$, and performed a least-squares fit to the frequencies as a function of mode order $n$, where the slope of this fit is $\Delta v$ (ref. 43). Small separations $\delta v_{0,2}$ are calculated weighting $v_{0,n} − v_{2,n-1}$ by the same Gaussian window as before, now performing a least-squares fit to $v_{2,n-1} − v_{max}$, and extracting the intercept of the fit[43].

**Inner phase shifts.** We calculate the inner phase shift[44] of a particular mode, $\phi_\ell$, as a function of the acoustic radius, $t = \int_0^r dr/c_s(r)$, by evaluating

$$\phi_\ell(t) = \tan^{-1}\left(\frac{\omega\psi}{d\psi/dt}\right) - \omega t + \frac{\pi}{2}\ell \tag{1}$$

at location $t = 0.5T$, for both the radial ($\ell = 0$) and quadrupole ($\ell = 2$) modes, where $T$ is the acoustic radius at the surface of the model, $\omega$ is the angular mode frequency and $\psi = rp'/\sqrt{c_s\rho}$, with $r$ being the radius, $c_s$ the speed of sound, $\rho$ the density and $p'$ the Eulerian pressure perturbation of the mode. For each stellar model and degree, we evaluate the inner phase shifts for all modes[3], then perform a weighted average over the frequencies using a Gaussian window[4] centred at $v_{max}$ with a full-width at half-maximum of[45]

$$\Gamma = 0.66\ \mu Hz \times (v_{max}(\mu Hz))^{0.88}. \tag{2}$$

**Density kernel.** Sharply localized structural features in the stellar interior perturb $p$-mode frequencies from the uniform spacing predicted by their asymptotic relation. For $p$-modes in particular, features in the density and the speed of sound in particular yield such frequency perturbations through integrals against localization kernels:

$$\frac{\delta\omega_i}{\omega_i} \sim \int K_{\rho,c_s^2,i}\frac{\delta\rho}{\rho}\,dr + \int K_{c_s^2,\rho,i}\frac{\delta c_s^2}{c_s^2}\,dr, \tag{3}$$

where $\delta\rho$ and $\delta c_s^2$ indicate departures in the density and sound-speed profiles of the star from, say, a smoothly stratified polytrope. We compute these as[46]

$$\begin{aligned}
K_{c_s^2,\rho}(r) &= \frac{\rho c_s^2\chi^2 r^2}{2I\omega^2}; \\
K_{\rho,c_s^2}(r) &= \frac{\rho r^2}{2I\omega^2}\Bigg[c_s^2\chi^2 - \omega^2(\xi_r^2 + \Lambda\xi_h^2) - 2g\xi_r\chi \\
&\quad - 4\pi G\int_r^R \xi_r\left(2\rho\chi + \xi_r\frac{d\rho}{dr}\right)dr' \\
&\quad + 2g\xi_r\frac{d\xi_r}{dr} + 4\pi G\rho\xi_r^2 + 2\left(\xi\frac{d\Phi}{dr} + \Lambda\xi_h\frac{\Phi}{r}\right)\Bigg],
\end{aligned} \tag{4}$$

where $\xi_r$ and $\xi_h$ are the radial and horizontal components of the Lagrangian displacement $\xi$ of the mode, $\chi = (\nabla \cdot \xi)/Y_\ell^m$, $\Lambda = \ell(\ell + 1)$, $g = Gm/r^2$ is the local gravitational field and $\Phi$ is the perturbation to the gravitational potential. These two kernels are offset from each other by a phase lag of $\pi/2$. The averaged kernel shown in Fig. 3 was then constructed by averaging $K_{\rho,c_s^2}$ over all radial orders near $v_{max}$ with weights given by a Gaussian envelope centred on $v_{max}$ with width given by equation (2).

## Data availability

K2 light curves are available from the Mikulski Archive for Space Telescopes (https://archive.stsci.edu/). Power spectra, isochrone and stellar tracks are available at Zenodo (https://doi.org/10.5281/zenodo.12617071; ref. 49).

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

**Acknowledgements** D.S. is supported by the Australian Research Council (DP190100666). J.O. acknowledges support from NASA through the NASA Hubble Fellowship grant HST-HF2-51517.001, awarded by STScI. STScI is operated by the Association of Universities for Research in Astronomy, Incorporated, under NASA contract NAS5-26555.

**Author contributions** C.R. computed the power spectra, measured the frequency separations and was the primary writer of the manuscript; J.O. and C.L. calculated the inner phase shifts and kernel functions; C.R. and C.L. computed the theoretical models; D.S. and C.R. performed the analysis; C.R., D.S., J.O., C.L., T.R.B. and M.H. collaboratively discussed the results and contributed to writing of the paper.

**Funding** Open access funding provided through UNSW Library.

**Competing interests** The authors declare no competing interests.

**Additional information**
**Correspondence and requests for materials** should be addressed to Claudia Reyes.

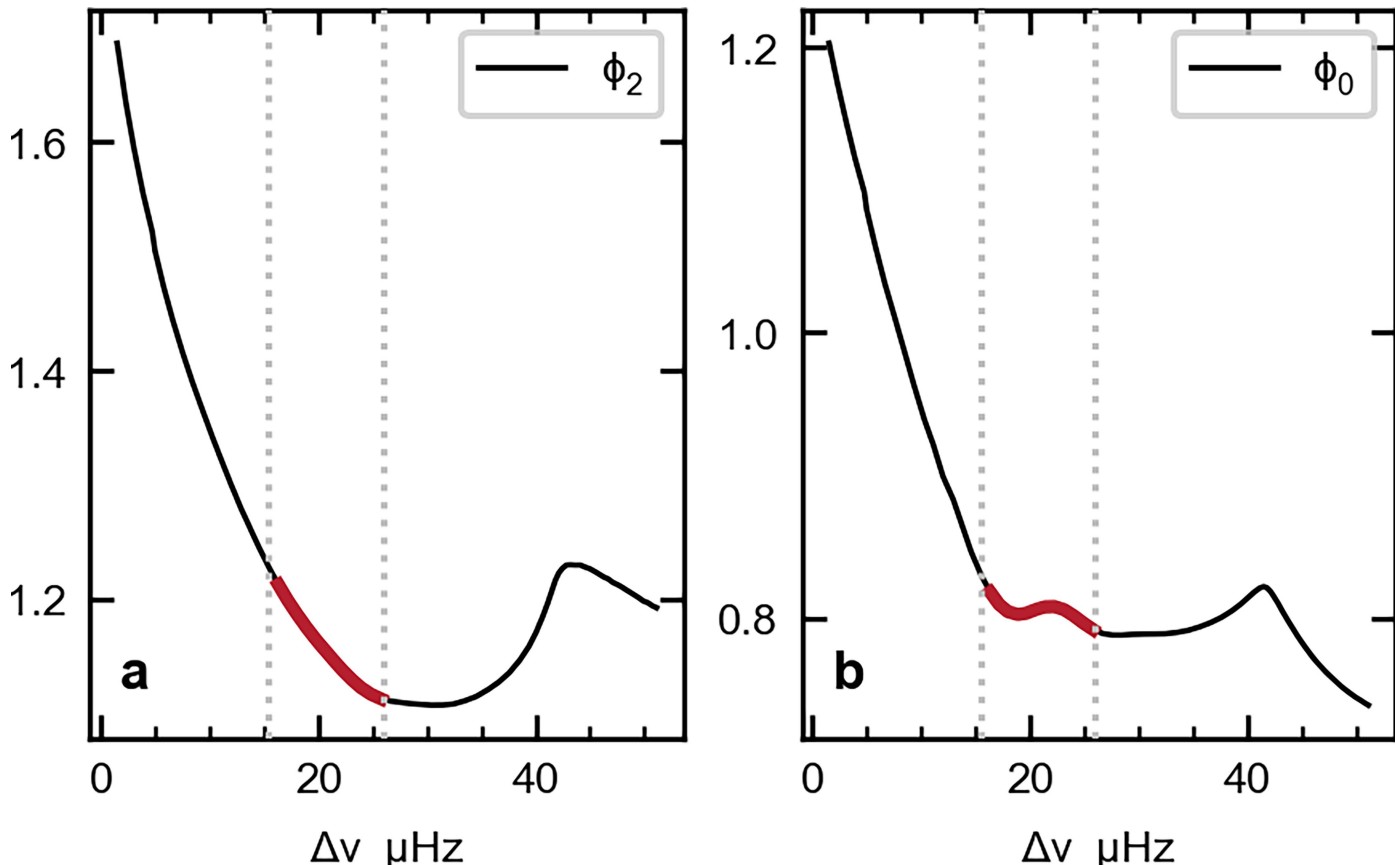

**Extended Data Fig. 1 | Inner phase shifts in isochrone models.** Inner phase shifts $\phi_0$ and $\phi_2$ as averaged over modes near $\nu_{max}$ from models along the M67 isochrone[22]. The dotted lines indicate the $\Delta\nu$ boundaries of Fig. 3e. Although no change in slope was observed in $\phi_2$ at the frequencies of the feature (a), a change in slope was found in the evolution of $\phi_0$ (b).

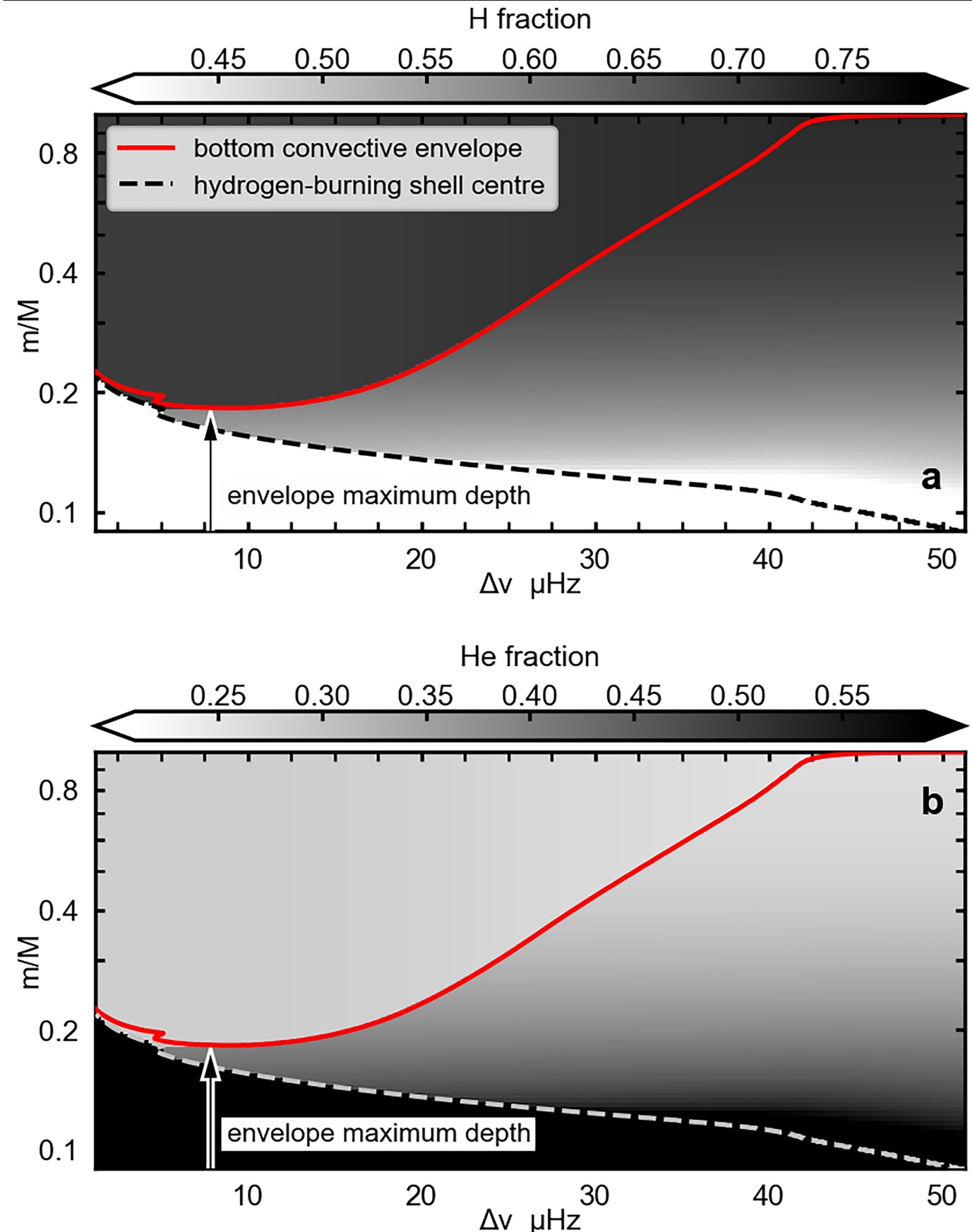

**Extended Data Fig. 2 | Hydrogen and helium fractions and the bottom of the convective envelope.** Figures are equivalent to Fig. 3d, but with a grey scale showing the hydrogen and helium fractions. The bottom of the convective envelope (in red), traces the chemical discontinuity that leads to large gradients in molecular weight and temperature, as described in the main text.

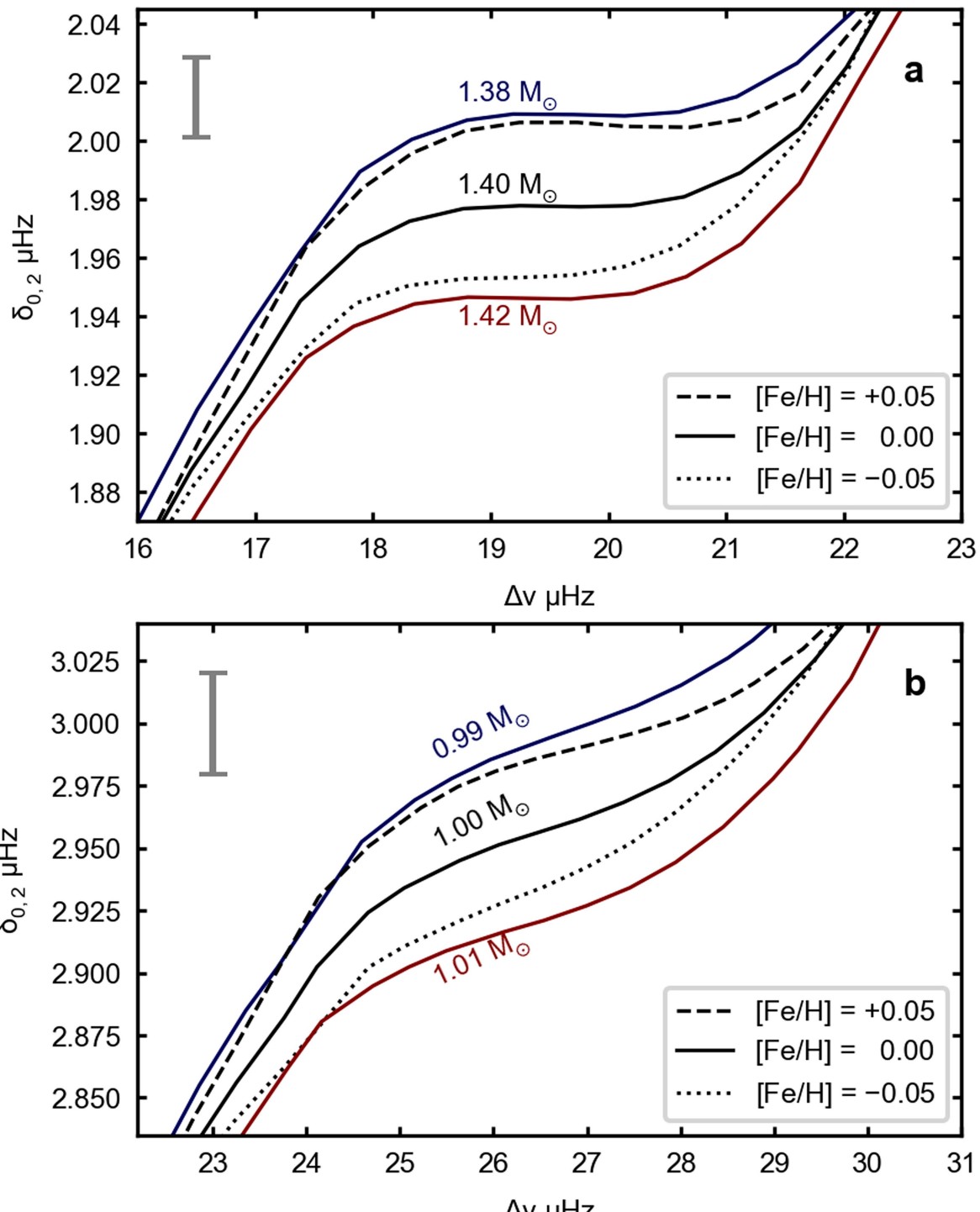

**Extended Data Fig. 3 | Metallicity sensitivity of plateau features.** All solid lines correspond to [Fe/H] = 0 tracks of masses as annotated. The dashed and dotted lines correspond to metallicity variations of the central track, of mass 1.4 M$_\odot$, or 1.0 M$_\odot$, as per the legend. The figures illustrate that translating a typical [Fe/H] uncertainty, estimated at 0.05 to 0.10 dex[47,48], into the plateau indicates that for solar-metallicity tracks at 1.40 M$_\odot$, this uncertainty corresponds to less than 0.02 M$_\odot$. For solar-metallicity tracks at 1.00 M$_\odot$, the [Fe/H] uncertainty translates to less than a 0.01 M$_\odot$ uncertainty. Typical *Kepler* error bars for small separations of 2 µHz and 3 µHz are shown next to the 1.4 M$_\odot$ and 1.0 M$_\odot$ tracks, respectively (see Methods).

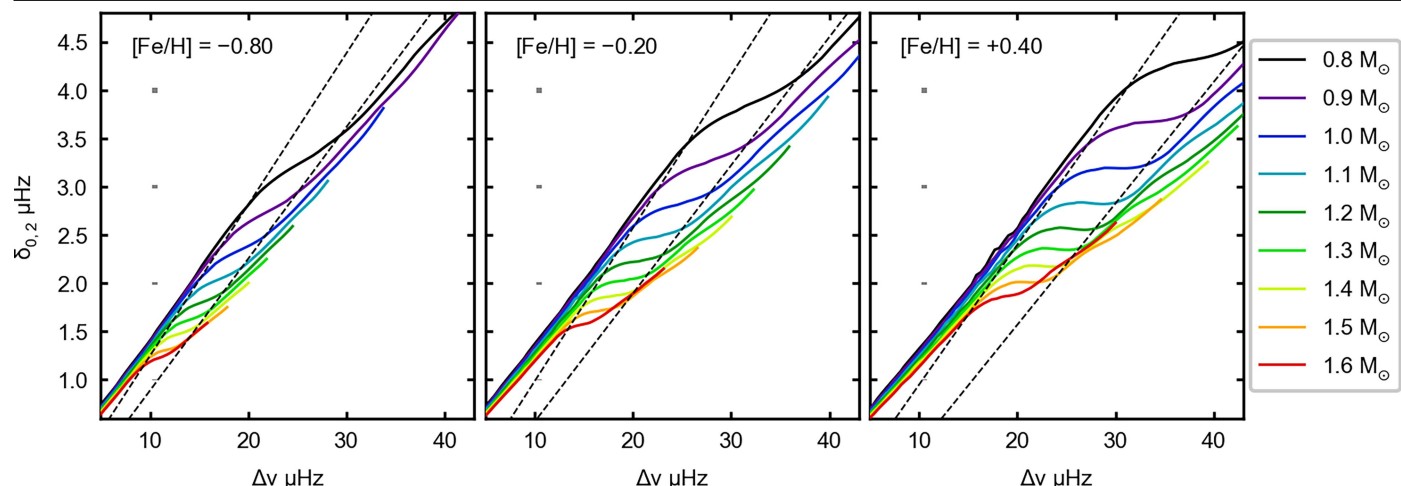

**Extended Data Fig. 4 | Plateau-like features across different metallicities.** The figures illustrate the metallicity dependence of plateau-like features across [Fe/H] from −0.8 to +0.4 dex, showing masses from 0.8 M$_\odot$ to 1.6 M$_\odot$. Dotted lines approximately trace the region where we see the plateau for each metallicity. The figure illustrates the shift of the plateaus from the lower left to the upper right corner of the diagram, with the progression from the most metal-poor to the most metal-rich tracks. For the latter, flatter and wider plateaus emerge. Typical *Kepler* uncertainties (see Methods) are indicated at $\Delta v = 10$ for two = 2, 3, and 4 $\mu$Hz.

**Extended Data Table 1 | Seismic parameters of M67 stars**

| EPIC ID | $\Delta\nu$ | $\sigma_{\Delta\nu}$ | $\delta\nu_{0,2}$ | $\sigma_{\delta\nu_{0,2}}$ |
|---|---|---|---|---|
| 211407537 | 1.33 | 0.05 | 0.23 | 0.00 |
| 211380313 | 2.14 | 0.03 | 0.36 | 0.05 |
| 211410817 | 2.56 | 0.02 | 0.38 | 0.02 |
| 211406541 | 4.29 | 0.07 | 0.57 | 0.15 |
| 211392837 | 4.85 | 0.02 | 0.71 | 0.10 |
| 211413623 | 6.32 | 0.02 | 0.86 | 0.03 |
| 211396385 | 7.03 | 0.02 | 0.84 | 0.03 |
| 211414300 | 7.12 | 0.04 | 0.93 | 0.02 |
| 211408346 | 8.21 | 0.02 | 0.89 | 0.10 |
| 211410231 | 8.93 | 0.02 | 1.09 | 0.04 |
| 211412928 | 9.74 | 0.02 | 1.19 | 0.05 |
| 211384259 | 13.83 | 0.04 | 1.74 | 0.13 |
| 211411629 | 14.34 | 0.02 | 1.68 | 0.23 |
| 211406144 | 14.50 | 0.03 | 1.83 | 0.11 |
| 211414687 | 15.25 | 0.03 | 1.82 | 0.04 |
| 211416749 | 16.95 | 0.04 | 1.99 | 0.08 |
| 211421954 | 17.62 | 0.20 | 2.05 | 0.08 |
| 211409560 | 19.17 | 0.04 | 2.07 | 0.03 |
| 211388537 | 20.87 | 0.26 | 2.05 | 0.04 |
| 211403248 | 21.46 | 0.22 | 1.96 | 0.21 |
| 211405262 | 27.08 | 0.29 | 2.55 | 0.11 |
| 211415364 | 28.47 | 0.02 | 2.74 | 0.20 |
| 211413064 | 29.16 | 0.81 | 2.59 | 0.08 |
| 211409088 | 32.99 | 0.02 | 2.88 | 0.14 |
| 211411922 | 33.98 | 0.24 | 2.73 | 0.13 |
| 211414203 | 38.53 | 0.27 | 3.33 | 0.14 |
| 211407836 | 39.02 | 0.12 | 3.72 | 0.12 |

Stars from M67 with measured small frequency separation $\delta\nu_{0,2}$. The uncertainty $\sigma_{\Delta\nu}$ is taken from results by the pysyd pipeline. $\Delta\nu$, $\delta\nu_{0,2}$, and $\sigma_{\delta\nu_{0,2}}$ are calculated as described in Methods. All values are given in $\mu$Hz.