## [Peer Review File · Nature]

Acoustic modes in M67 cluster stars trace deepening convective envelopes

Corresponding Author: Dr Claudia Reyes

Version 1:

Reviewer comments:

Referee #1

(Remarks to the Author)

The context

The context is the study of the stellar clusters as a powerful tool for validating the theoretical description of stars. The assumption of a similar age, distance, and chemical composition might provide constraints on each cluster's member, helping to significantly improve our understanding of stellar evolution. In the present manuscript, the authors apply asteroseismic techniques to study 27 stars belonging to the well-studied open cluster M67 (NGC2682) by using high-quality data obtained by the NASA K2 space mission (Howell et al. 2014).

The abstract

The abstract appears clear: the authors claim to be able with their analysis to detect internal structure changes as the stars evolve off the main sequence phase and the convective envelopes get deeper. In addition, they plan to show the connection between the seismic parameters and the internal chemical discontinuities and to be able to improve estimations of mass and age for this cluster.

Conclusion and clarity

The manuscript, in my opinion, is not adequate for general scientific readers (e.g., physical parameters and equations are not adequately introduced, general characteristics of solar-like stars are not described, the difference between radial and non-radial modes is not clear, the definition of overshooting is not given etc). In general, the authors have probably not succeeded in pointing out what is the importance of the present results for general purposes.

At the same time, the content needs some further discussions to be of interest to researchers working in the field. I can see a general lack of details on the results.

In addition, it turns out to be obscure how all of this should improve the estimation of age and mass, in comparison to previous results, as announced in the abstract.

I suggest the authors take the opportunity to better explore the potential of their results, making an effort to include more details of their analysis and results and in particular to discuss them.

This appears more evident perhaps in the conclusion which leaves the reader with a series of unresolved questions: what do you mean by the influence of a deep convective envelope, what is an ultra-deep regime of the convective envelope and what do you mean for

'leaving a chemical discontinuity imprint'. No figures show the changes in the abundances H or He abundances, which might explain what is happening from the chemical point of view or figures showing the variation of the molecular weight. Panels of Figure S2 which look very interesting to me have not been commented on and might help the authors write the conclusion.

Originality

The inferred open cluster M67 has already been subject to intensive ground-based and space-based campaigns. In particular asteroseismic analysis has been produced previously for targets belonging to this cluster leading previous authors to estimate the age and the mass of the studied sample (Gilliland et al. 1993, Stello et al. 2016, Li et al. 2024).

However, the manuscript appears to have some originality.

Here the authors present for the first time the analysis of high-precision photometric data of 27 stellar targets belonging to the

same cluster as detected by the K2 space mission. The quality of the obtained data allows the authors to detect in the oscillation spectra not only the large separation but also the small separations and to build the so-called CD diagram, which like the HR diagram allows to represent the evolutionary history of the considered population in a two-dimensional plan. Figure 2, although a bit crowded, is very interesting and shows most of the results.

Isochrone fitting of the observed stars has been well developed by previous authors (e.g., Viani & Basu 2017 and Souto et al. 2019) who considered all the possible additional effects such as mixing, overshooting, diffusion and so on. Viani & Basu, in particular, have already obtained a similar conclusion on the occurrence of envelope overshooting. Nevertheless, the results presented if better detailed might interest many people working in stellar physics.

I
- Data & methodology and statistics

The authors use a well-developed strategy for the analysis of photometric data and for extracting oscillation spectra features. In particular, they use some asteroseismic tools, such as the phase shifts and the kernel density to explain what is happening in the interior.

For the sake of clarity, Table 1 should list all the spectroscopic parameters such as effective temperature and gravity and all the obtained seismic parameters such as individual ages, stellar mass and radius and in particular depth of the convective region.

The results and data shown in the tables need to be properly commented on.

-Figures and tables

All error bars should be defined in the corresponding figure legends or the caption.

Figure 2 reports only errors in the small separations and not the errors in large separations.

The caption should adequately explain Figure S2.

-References.

In my opinion, most of the important bibliography is missing here, in particular, that related to important aspects such as the K2 space photometric mission, the results already obtained on the cluster under study or different ones, the variation of the large and small separations with age and in particular in the most evolved phases, the use of CD diagram. Some suggestions for additional references are given above but many others are missing.

To conclude, I think that the perspectives of the present manuscript might be relevant to the broad field of stellar physics, but, unfortunately in the actual state, it is not yet ready for publication. The improvements suggested above and the comparison of the results with the previous ones can certainly enhance the interest.

Referee #2

(Remarks to the Author)

The authors use K2 light curves for M67 cluster members to study the evolution of the small frequency separations as the stars evolve from the sub-giant phase towards and up the red-giant branch. While doing so, they provide strong evidence for a temporary stalling of the evolution of these differences that can be traced back to the deepening of the convective envelope and the radial-modes sensitivity to the structural glitch at the transition to the radiative core. The interpretation of the observational evidence is strongly supported by the analysis of pure acoustic mode pulsations computed from stellar models using a novel technique developed earlier by one of the co-authors.

This is a work of significant impact in the field, showing for the first time the potential of using the small separations as means to constrain masses (hence ages) of evolved stars, despite the earlier apparent degeneracy of these differences in the so-called C-D diagram. I do not find any flaws or obvious weaknesses in the work presented. In my opinion, the manuscript is very clearly written (both the main text and the Methods sections), providing enough details to be followed by experts in the field while still appealing to the broader scientific community. For these reasons, I recommend it for publication in Nature.

The errors on the observational quantities are shown/provided in the relevant figures/tables (albeit not described in the figure captions, as per Nature guidelines to referees) but their computation is explained in the Methods section.

I have only a couple of comments for the authors to consider expanding on so as to make their conclusions more robust, as well as a few typos noted below.

-- the authors mention that "this new diagnostic tool could be used to estimate the masses of field stars". Yet, they do not at any moment discuss the impact of the uncertainty in metallicity on such potential inferences. Given that metallicity impacts the opacity, thus, the depth of stellar convective envelopes, did the authors attempt to quantify the mass-metallicity degeneracy of the observed phenomena and the consequent limitation to the mass inferences on field stars they suggest?

--Likewise, while the potential of using the observed feature in clusters to constrain the amount of overshoot is very interesting, I wonder to what degree uncertainties in the metallicity and other aspects of the physics that may influence the isochrone choice for the cluster could impact such inferences. Did the authors propagate these uncertainties to their model results to understand how they may impact the potential of using this feature to infer on overshoot?

Minor issues:

line 28: what is ν in the integral definition of the small separation?

Figure 2, panels a) and b): Y label (ν is missing)

Figure 3: please consider replacing the word “discontinuity”. A discontinuity in the evolution would imply that the derivative of the phase with respect the large frequency separation would jump at one given point, which I believe from the plot not to be the case.

Figure 4, panel b): either connect the inset to the appropriate region in the main plot or provide values for the axes of the inset (or both).

Methods (Density Kernel), line 199: “over all modes”. In the main text it is indicated that only kernels for the radial modes are considered. The fact that the general expression for the kernels is provided in this section and that the authors say “over all modes” can confuse the reader. If the kernel amplitude shown in Figure 2 is computed from the radial modes around ν_{\max} , then I suggest being explicit about that here too.

Figure S1, caption: be specific about the metallicity used.

Margarida Cunha

Version 2:

Reviewer comments:

Referee #1

(Remarks to the Author)

The revised version of the manuscript has been improved as the authors have satisfactorily addressed all comments and questions in my first report.

Let me point out that my criticism was raised to strengthen the article in order to meet the publication criteria of the present journal.

I think the article deserves to be published in the journal: the methods are well described and the results are really important for the field.

Nevertheless, I still have a few minor questions that might require some attention from the authors, if wished:

1) Page 7: It might be worth mentioning, although out of the scope, that the location of the convective boundary might be affected by other kinds of additional effects: e.g. rotational mixing, gravitational settling, diffusion, or something else occurring or competing at the base of the convective envelope.

2) In the insight of Figure 4: Similar red colors are used to distinguish different curves. Different palettes or different line types might be introduced.

3) Line 135: the word 'dynamics' might appear slightly misleading since usually it refers to rotational motion.

Referee #2

(Remarks to the Author)

I thank the authors for their additional effort to clarify the points I raised in my first report. I find the changes made to the manuscript (also in reply to the other referee) adequate and recommend the manuscript for publication.

Margarida Cunha

Referee #1 (Remarks to the Author):

The abstract

The referee writes “to be able to improve estimations of mass and age for this cluster”.

We have clarified that this is in regards to field stars, in blue text, line 17.

Conclusion and clarity

The referee writes “physical parameters and equations are not adequately introduced, general characteristics of solar-like stars are not described, the difference between radial and non-radial modes is not clear, the definition of overshooting is not given etc) “

- In lines 21-25 there was a description of how, for solar-like stars, the waves in radial and non-radial modes travel, and that non-radial modes “are confined between an inner turning point and the surface”. We have now also added that the oscillations in these solar-like stars are excited by convection (line 21).
- In lines 32-43 we gave a description of solar-like stars and how their oscillations evolve.
- In lines 79-80 we now reiterate the key difference relevant for the findings in this paper, that radial modes are the only p-modes allowed in the deepest parts of stars “... lying near the star’s centre, where only radial ($\ell=0$) p-modes reach, beyond the inner turning point of $\ell=2$ modes”,
- We have introduced the concept of overshooting. See lines 98-103.

The referee writes “the content needs some further discussions to be of interest to researchers working in the field. I can see a general lack of details on the results. In addition, it turns out to be obscure how all of this should improve the estimation of age and mass, in comparison to previous results, as announced in the abstract.”

We believe this comment might be due to a misinterpretation of the aim and result of this study. We believe we have now succeeded in clarifying that M67 acts as the observational test-bench (to reveal the presence of our newly discovered plateau in the CD diagram) and not the final target for improved stellar properties of the cluster itself.

The referee writes “This appears more evident perhaps in the conclusion which leaves the reader with a series of unresolved questions: what do you mean by the influence of a deep convective envelope, what is an ultra-deep regime of the convective envelope and what do you mean for 'leaving a chemical discontinuity imprint'. No figures show the changes in the abundances H or He abundances, which might explain what is happening from the chemical point of view or figures showing the variation of the molecular weight. Panels of Figure S2 which look very interesting to me have not been commented on and might help the authors write the conclusion.”

- We have replaced “the influence of the convective envelope” with “the depths reached by convective envelopes lead to measurable effects in the small-separation frequencies”, lines 137-138.
- We better define what we mean by an ultra-deep regime: “... beginning when roughly 80% of the star’s mass is undergoing convection.”, line 139-140.
- We believe the new figure: Extended Data Figure 2, helps to explain the chemical discontinuity imprint, which we also refer to in lines 84-86 when we say “Large density and sound-speed gradients are known to exist at such boundaries due to differing chemical compositions on either side”
- Part of the restructuring of this paper was to bring figure S2 to the main article, now Figure 2. We have included a new subsection describing this figure named “Small frequency separations follow the structure of nuclear burning zones”, lines 45-53.

Originality

The referee writes “The inferred open cluster M67 has already been subject to intensive ground-based and spacebased campaigns. In particular asteroseismic analysis has been produced previously for targets belonging to this cluster leading previous authors to estimate the age and the mass of the studied sample (Gilliland et al. 1993, Stello et al. 2016, Li et al. 2024).”

- We agree. However, as mentioned the cluster is our tool, not the target. Still, we take the opportunity to follow this suggestion when we specify that we take our lightcurves from the Kepler/K2 mission (lines 57-58).

The referee writes “The quality of the obtained data allows the authors to detect in the oscillation spectra not only the large separation but also the small separations and to build the so-called CD diagram, which like the HR diagram allows to represent the evolutionary history of the considered population in a two-dimensional plan.”

- We believe our changes to the manuscript help clarify that, importantly, the main result reported here is a new feature in the CD diagram that does NOT have a counterpart in the HR diagram

The referee writes “Isochrone fitting of the observed stars has been well developed by previous authors (e.g., Viani & Basu 2017 and Souto et al. 2019) who considered all the possible additional effects such as mixing, overshooting, diffusion and so on. Viani & Basu, in particular, have already obtained a similar conclusion on the occurrence of envelope overshooting. Nevertheless, the results presented if better detailed might interest many people working in stellar physics.”

- We do not perform isochrone fitting but merely adopt an existing isochrone from Reyes et al 2024 in which comparisons and discussions related to previous results are

included, we have tried to clarify this in lines 62-63. Notably the models we use for this study adopted a solar calibration of envelope overshooting by Choi, 2016, line 105.

Data & methodology and statistics

The referee writes “For the sake of clarity, Table 1 should list all the spectroscopic parameters such as effective temperature and gravity and all the obtained seismic parameters such as individual ages, stellar mass and radius and in particular depth of the convective region.”

- We do not derive nor use any of this in our work. We believe that now that the aim of our work is more clear, the referee will agree that this does not pertain to our study.

Figures and tables

The referee writes “All error bars should be defined in the corresponding figure legends or the caption. Figure 2 reports only errors in the small separations and not the errors in large separations. The caption should adequately explain Figure S2.”

- In the caption for the old Figure 2, now Figure 3 we have now specified that large-separation error bars are smaller than the symbols (and the error is negligible), and that δ_{02} error bars were obtained as detailed in Methods.
- The caption for the old Figure S2, now Figure 2 has been modified, and the new subsection “Small frequency separations follow the structure of nuclear burning zones” has been added to better explain the figure.

References

The referee writes “In my opinion, most of the important bibliography is missing here, in particular, that related to important aspects such as the K2 space photometric mission, the results already obtained on the cluster under study or different ones, the variation of the large and small separations with age and in particular in the most evolved phases, the use of CD diagram. “

- We have added lines 56-58 “This cluster ... population, which have been the target of seismic studies for decades (Gilliland 1993). Recent work (Stello, 2016) includes a study of its giants using Kepler/K2 data (Howell, 2014), which we also use in this study”.
- In lines 33-34, we note that “in main sequence stars, small separations are a good indicator of evolutionary state” (Christensen-Dalsgaard, 1988). We then explain that beyond the main sequence, variations in small-frequency separations are harder to interpret due to the scatter of mixed modes. However, as coupling weakens (with

mixed modes also weakening), small-frequency separations become nearly proportional to large-frequency separations (Montalban, 2010; Huber, 2010). All this was in the original version, except for the Montalban reference which is a recent addition we found relevant for this context.

The referee writes “... the comparison of the results with the previous ones can certainly enhance the interest.”

- Similarly as before, we believe that now that the aim and result of our work has been made more clear, the referee will agree that the comment is moot.

Referee #2 (Remarks to the Author):

The referee writes “-- the authors mention that “this new diagnostic tool could be used to estimate the masses of field stars”. Yet, they do not at any moment discuss the impact of the uncertainty in metallicity on such potential inferences. Given that metallicity impacts the opacity, thus, the depth of stellar convective envelopes, did the authors attempt to quantify the mass-metallicity degeneracy of the observed phenomena and the consequent limitation to the mass inferences on field stars they suggest?”

And “Likewise, while the potential of using the observed feature in clusters to constrain the amount of overshoot is very interesting, I wonder to what degree uncertainties in the metallicity and other aspects of the physics that may influence the isochrone choice for the cluster could impact such inferences. Did the authors propagate these uncertainties to their model results to understand how they may impact the potential of using this feature to infer on overshoot? “

- To address these comments we have added two new figures: Extended Data Figures 3 and 4, which we refer to in line 132.

Minor issues

The referee writes:

line 28: what is ν in the integral definition of the small separation?

- We added in line 31 “for a given frequency ν ”, as this expression is frequency dependant, and to obtain a representative value of the small frequency separation the values should be averaged in frequency using a gaussian window centred at ν_{\max} [Equation (2)], a similar approach to the one used for the phase shifts (starting line 206), and kernel (starting in line 214)

Figure 2, panels a) and b): Y label (ν is missing)

- The old Figure 2 is now Figure 3: $\Delta_{0,2} \mu \text{ Hz}$ is present on the left exterior side of panel a), and on the left, interior side of panel b). Furthermore, we now explicitly mention that the x-axis is shared between panels a) and d), which is why we only write the x label at the bottom.

Figure 3: please consider replacing the word “discontinuity”. A discontinuity in the evolution would imply that the derivative of the phase with respect the large frequency separation would jump at one given point, which I believe from the plot not to be the case.

- Old Figure 3 is now Extended Data Figure 1, and we have taken the referee’s suggestion, and replaced “discontinuity” here with “a change in slope”.

Figure 4, panel b): either connect the inset to the appropriate region in the main plot or provide values for the axes of the inset (or both).

- We have now connected the inset and added values to inset axes in Figure 4b.

Methods (Density Kernel), line 199: “over all modes”. In the main text it is indicated that only kernels for the radial modes are considered. The fact that the general expression for the kernels is provided in this section and that the authors say “over all modes” can confuse the reader. If the kernel amplitude shown in Figure 2 is computed from the radial modes around ν_{\max} , then I suggest being explicit about that here too.

- This was a mistake, it is supposed to read “over all radial orders”, now fixed. We have also specified now that this refers only to “radial modes” in line 91.

Figure S1, caption: be specific about the metallicity used.

- Figure S1 is now Figure 2, and we write in the caption and in the text that these are solar metallicity tracks.

Response to remaining issues raised by referees.

I would like to thank both referees for their valuable feedback and suggestions throughout the review process. Below, I address the remaining points raised by Referee #1 and outline the corresponding revisions made to the manuscript.

1. **Page 7 – Additional effects influencing the convective boundary:**
I was unable to incorporate this explanation without disrupting the overall flow of the text. However, I have ensured that the manuscript does not imply overshooting is the sole factor influencing the location of the convective boundary.
2. **Figure 4 – Improved color scheme:**
I have updated the color scheme to make the curves easier to distinguish.
3. **Line 135 – Revised section title:**
The section title has been revised from “M67 reveals deep stellar dynamics” to “M67 reveals structural changes in deep stellar interiors” to avoid potential misinterpretation of the term "dynamics."

- Claudia Reyes